# Modeling Continuous Stochastic Processes with Dynamic Normalizing Flows

**Ruizhi Deng**[1,2*]   **Bo Chang**[1]   **Marcus A. Brubaker**[1,3,4]   **Greg Mori**[1,2]   **Andreas M. Lehrmann**[1]
[1]Borealis AI   [2]Simon Fraser University   [3]York University   [4]Vector Institute

## Abstract

Normalizing flows transform a simple base distribution into a complex target distribution and have proved to be powerful models for data generation and density estimation. In this work, we propose a novel type of normalizing flow driven by a differential deformation of the Wiener process. As a result, we obtain a rich time series model whose observable process inherits many of the appealing properties of its base process, such as efficient computation of likelihoods and marginals. Furthermore, our continuous treatment provides a natural framework for irregular time series with an independent arrival process, including straightforward interpolation. We illustrate the desirable properties of the proposed model on popular stochastic processes and demonstrate its superior flexibility to variational RNN and latent ODE baselines in a series of experiments on synthetic and real-world data.

## 1   Introduction

Expressive models for sequential data form the statistical basis for downstream tasks in a wide range of domains, including computer vision, robotics, and finance. Recent advances in deep generative architectures, especially the concept of reversibility, have led to tremendous progress in this area and created a new perspective on many of the long-standing limitations that are typical in traditional approaches based on structured decompositions (e.g., state-space models).

We argue that the power of a time series model depends on its properties in the following areas: (1 – Resolution) Common time series models are discrete with respect to time. As a result, they make the implicit assumption of a uniformly spaced temporal grid, which precludes their application from asynchronous tasks with a separate arrival process. (2 – Structural assumptions) The expressiveness of a temporal model is determined by the dependencies and shapes of its variables. In particular, the topological structure should be rich enough to capture the dynamics of the underlying process but sparse enough to allow for robust learning and efficient inference. (3 – Generation) A good time series model must be able to generate unbiased samples from the true underlying process in an efficient way. (4 – Inference) Given a trained model, it should support standard inference tasks, such as interpolation, forecasting, and likelihood calculation.

Recently, deep generative modeling has enabled vastly increased flexibility while keeping generation and inference tractable, owing to novel techniques like amortized variational inference [28, 12], reversible generative models [41, 29], and networks based on differential equations [9, 34].

In this work, we approach the modeling of continuous and irregular time series with a *reversible generative model for stochastic processes*. Our approach builds upon ideas from normalizing flows; however, instead of a static base distribution, we transform a dynamic base process into an observable one. In particular, we introduce the *continuous-time flow process (CTFP)*, a novel type of generative model

that decodes the base continuous Wiener process into a complex observable process using a dynamic instance of normalizing flows. The resulting observable process is thus continuous in time. In addition to the appealing properties of static normalizing flows (e.g., efficient sampling and exact likelihood), this also enables a series of inference tasks that are typically unattainable in time series models with complex dynamics, such as interpolation and extrapolation at arbitrary timestamps. Furthermore, to overcome the simple covariance structure of the Wiener process, we augment the reversible mapping with latent variables and optimize this latent CTFP variant using variational optimization. Our approach is illustrated in Figure 1.

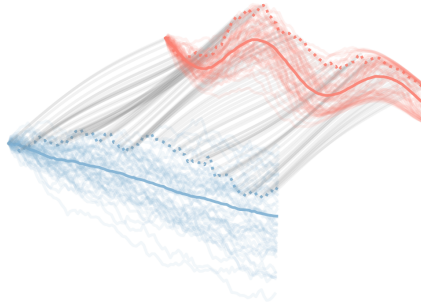

Figure 1: **Overview.** Wiener processes are continuous stochastic processes with appealing properties but limited flexibility. We propose to learn a complex observed process (red) through a differential deformation (grey) of the base Wiener process (blue), thereby preserving the advantages of the base process.

**Contributions.** In summary, we propose the *continuous-time flow process (CTFP)*, a novel generative model for continuous stochastic processes. It has the following appealing properties: (1) it induces flexible and consistent joint distributions on arbitrary and irregular time grids, with easy-to-compute density and an efficient sampling procedure; (2) the stochastic process generated by CTFP is guaranteed to have continuous sample paths, making it a natural fit for data with continuously-changing dynamics; (3) CTFP can perform interpolation and extrapolation conditioned on given observations. We validate our model and its latent variant on various stochastic processes and real-world datasets and show superior performance to state-of-the-art methods, including the variational recurrent neural network (VRNN) [12] and latent ordinary differential equation (latent ODE) [42].

## 2    Related Work

The following sections discuss the relevant literature on statistical models for sequential data and put it in context with the proposed approach.

**Early Work.** Among the most popular traditional time series models are latent variable models following the state-space equations [16], including the well-known variants with discrete and linear state-space [2, 26]. In the non-linear case, exact inference is typically intractable and we need to resort to approximate techniques [25, 23, 7, 8, 45]. Our CTFP can be viewed as a form of a continuous-time extended Kalman filter where the nonlinear observation process is noiseless and invertible and the temporal dynamics are a Wiener process. The final result, however, is more expressive than a Wiener process but retains some of its appealing properties like closed-form likelihood, interpolation, and extrapolation. Tree-based variants of non-linear Markov models have been proposed in [33]. An augmentation with switching states increases the expressiveness of state-space models but introduces additional challenges for learning [17] and inference [1]. Marginalization over an expansion of the state-space equations in terms of non-linear basis functions extends classical Gaussian processes [40] to Gaussian process dynamical models [24].

**Variational Sequence Models.** Following its success on image data, many works extended the variational autoencoder (VAE) [28] to sequential data [5, 12, 18, 35]. While RNN-based variational sequence models [12, 5] can model distributions over irregular timestamps, those timestamps have to be discrete and thus the models lack the notion of continuity. As a result, they are not suitable for modeling sequential data that have continuous underlying dynamics. Furthermore, it is not straightforward to perform interpolation at arbitrary timestamps using those models.

Latent ODEs [42] use an ODE-RNN as encoder and propagate a latent variable along a time interval using a neural ODE. This formulation ensures that the latent trajectory is continuous in time. However, decoding of the latent variables to observations is done at each time step independently. As a result, there is no guarantee that sample paths are continuous, which causes problems similar to the ones observed in variational sequence models. Neural stochastic differential equations (neural SDEs) [34] replace the deterministic latent trajectory of a latent ODE with a latent stochastic process but do also not generate continuous sample paths.

Recently, Qin et al. [39] proposed the recurrent neural process model. However, members of the neural process family [27, 19, 20, 44] only model the conditional distribution of data given observations and are not generic generative models.

**Normalizing Flows in Time Series Modeling.** Multiple recent works [32, 36, 43] apply reversible generative models to sequential data and show promise at capturing complex distributions. Mehrasa et al. [36] and Shchur et al. [43] use normalizing flows to model the distribution of inter-arrival time between events in temporal point processes. Kumar et al. [32] generate video frames using conditional normalizing flows. However, these models only use normalizing flows to model probability distributions in real space. In contrast, our model extends the domain of normalizing flows from distributions in real space to continuous-time stochastic processes.

## 3 Background

Our model is built upon the study of stochastic processes and recent advances in normalizing flow research. The following sections introduce the necessary background in these areas.

### 3.1 Stochastic Processes

A stochastic process can be defined as a collection of random variables that are indexed by time. An example of a continuous stochastic process is the *Wiener process*. The $d$-dimensional Wiener process $\boldsymbol{W}_\tau$ can be characterized by the following properties: (1) $\boldsymbol{W}_0 = 0$; (2) $\boldsymbol{W}_t - \boldsymbol{W}_s \sim \mathcal{N}(0, (t-s)\boldsymbol{I}_d)$ for $s \leq t$, and $\boldsymbol{W}_t - \boldsymbol{W}_s$ is independent of past values of $\boldsymbol{W}_{s'}$ for all $s' \leq s$. The joint density of $(\boldsymbol{W}_{\tau_1}, \ldots, \boldsymbol{W}_{\tau_n})$ can be written as the product of the conditional densities: $p_{(\boldsymbol{W}_{\tau_1}, \ldots, \boldsymbol{W}_{\tau_n})}(\boldsymbol{w}_{\tau_1}, \ldots, \boldsymbol{w}_{\tau_n}) = \prod_{i=1}^{n} p_{\boldsymbol{W}_{\tau_i}|\boldsymbol{W}_{\tau_{i-1}}}(\boldsymbol{w}_{\tau_i}|\boldsymbol{w}_{\tau_{i-1}})$ for $0 \leq \tau_1 < \cdots < \tau_n \leq T$.

The conditional distribution of $p_{\boldsymbol{W}_t|\boldsymbol{W}_s}$, for $s < t$, is multivariate Gaussian; its conditional density is

$$p_{\boldsymbol{W}_t|\boldsymbol{W}_s}(\boldsymbol{w}_t|\boldsymbol{w}_s) = \mathcal{N}(\boldsymbol{w}_t; \boldsymbol{w}_s, (t-s)\boldsymbol{I}_d), \tag{1}$$

where $\boldsymbol{I}_d$ is a $d$-dimensional identity matrix. This equation also provides a way to sample from $(\boldsymbol{W}_{\tau_1}, \ldots, \boldsymbol{W}_{\tau_n})$. Furthermore, given $\boldsymbol{W}_{t_1} = \boldsymbol{w}_{t_1}$ and $\boldsymbol{W}_{t_2} = \boldsymbol{w}_{t_2}$, the conditional distribution of $\boldsymbol{W}_t$ for $t_1 \leq t \leq t_2$ is also Gaussian:

$$p_{\boldsymbol{W}_t|\boldsymbol{W}_{t_1}, \boldsymbol{W}_{t_2}}(\boldsymbol{w}_t|\boldsymbol{w}_{t_1}, \boldsymbol{w}_{t_2}) = \mathcal{N}\left(\boldsymbol{w}_t; \boldsymbol{w}_{t_1} + \frac{t - t_1}{t_2 - t_1}(\boldsymbol{w}_{t_2} - \boldsymbol{w}_{t_1}), \frac{(t_2 - t)(t - t_1)}{t_2 - t_1}\boldsymbol{I}_d\right). \tag{2}$$

This is known as the Brownian bridge. An important property of the Wiener process is that the sample paths are continuous in time with probability one. This property allows our models to generate continuous sample paths and perform interpolation and extrapolation tasks.

### 3.2 Normalizing Flows

*Normalizing flows* [41, 13, 30, 14, 37, 29, 3, 10, 31, 38] are reversible generative models that allow both density estimation and sampling. If our interest is to estimate the density function $p_{\boldsymbol{X}}$ of a random vector $\boldsymbol{X} \in \mathbb{R}^d$, then normalizing flows assume $\boldsymbol{X} = f(\boldsymbol{Z})$, where $f : \mathbb{R}^d \to \mathbb{R}^d$ is a bijective function, and $\boldsymbol{Z} \in \mathbb{R}^d$ is a random vector with a simple density function $p_{\boldsymbol{Z}}$. The probability density function can be evaluated using the change of variables formula:

$$\log p_{\boldsymbol{X}}(\boldsymbol{x}) = \log p_{\boldsymbol{Z}}(g(\boldsymbol{x})) + \log\left|\det\left(\frac{\partial g}{\partial \boldsymbol{x}}\right)\right|, \tag{3}$$

where we denote the inverse of $f$ by $g$ and $\partial g/\partial \boldsymbol{x}$ is the Jacobian matrix of $g$. Sampling from $p_{\boldsymbol{X}}$ can be done by first drawing a sample from the simple distribution $\boldsymbol{z} \sim p_{\boldsymbol{Z}}$, and then apply the bijection $\boldsymbol{x} = f(\boldsymbol{z})$.

Chen et al. [9], Grathwohl et al. [21] proposed the *continuous normalizing flow*, which uses the *neural ordinary differential equation* (neural ODE) to model a flexible bijective mapping. Given $\boldsymbol{z} = \boldsymbol{h}(t_0)$ sampled from the base distribution $p_{\boldsymbol{Z}}$, it is mapped to $\boldsymbol{h}(t_1)$ based on the mapping defined by the ODE: $d\boldsymbol{h}(t)/dt = f(\boldsymbol{h}(t), t)$. The change in log-density is computed by the *instantaneous change of variables formula* [9]:

$$\log p_{\boldsymbol{X}}(\boldsymbol{h}(t_1)) = \log p_{\boldsymbol{Z}}(\boldsymbol{h}(t_0)) - \int_{t_0}^{t_1} \text{tr}\left(\frac{\partial f}{\partial \boldsymbol{h}(t)}\right) dt. \tag{4}$$

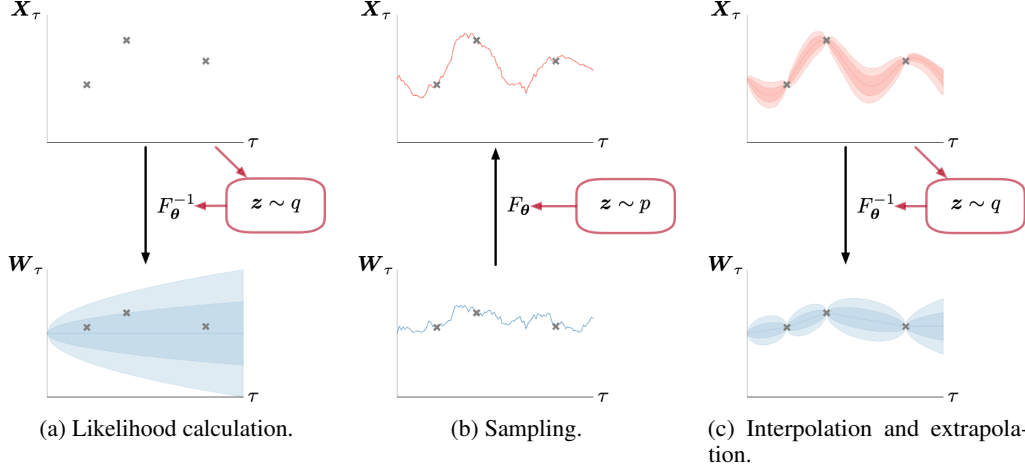

(a) Likelihood calculation.  (b) Sampling.  (c) Interpolation and extrapolation.

Figure 2: **(Latent) Continuous-Time Flow Processes (CTFPs).** (**a**) Likelihood calculation. Given an irregular time series $\{\boldsymbol{x}_{\tau_i}\}$, the inverse flow $F_{\boldsymbol{\theta}}^{-1}$ maps the observed process to a set of Wiener points $\{\boldsymbol{w}_{\tau_i}\}$ for which we can compute the likelihood according to Equation 7. (**b**) Sampling. Given a set of timestamps $\{\tau_i\}$, we sample a Wiener process and use the forward flow $F_{\boldsymbol{\theta}}$ to obtain a sample of the observed process. (**c**) Interpolation and extrapolation. In order to compute the density at an unobserved point $\boldsymbol{x}_\tau$, we compute the left-sided (extrapolation; Equation 1) or two-sided (interpolation; Equation 2) conditional density of its Wiener point $\boldsymbol{w}_\tau$ and adjust for the flow (Equation 11). **Notes:** The effect of the latent variables $\boldsymbol{Z}$ in our latent CTFP model is indicated by red boxes. The shaded areas represent 70% and 95% confidence intervals.

One potential disadvantage of the neural ODE model is that it preserves the topology of the input space, and there are classes of functions that cannot be represented by neural ODEs. Dupont et al. [15] proposed the augmented neural ODE (ANODE) model to address this limitation. Note that the original formulation of ANODE is not a generative model and it does not support the computation of likelihoods $p_{\boldsymbol{X}}(\boldsymbol{x})$ or sampling from the target distribution $\boldsymbol{x} \sim p_{\boldsymbol{X}}$. In this work, we formulate a modified version of ANODE that can be used as a (conditional) generative model.

# 4 Model

We define our proposed *continuous-time flow process (CTFP)* in Section 4.1. In Section 4.2, a generative variant of ANODE is presented as a component to implement CTFP. Since the proposed stochastic process is continuous in time, it enables interpolation and extrapolation at arbitrary time points, as described in Section 4.3. Finally, richer covariance structures are enabled by the latent CTFP model presented in Section 4.4.

## 4.1 Continuous-Time Flow Process

Let $\{(\boldsymbol{x}_{\tau_i}, \tau_i)\}_{i=1}^n$, denote a sequence of irregularly spaced time series data. We assume the time series to be an (incomplete) realization of a continuous stochastic process $\{\boldsymbol{X}_\tau\}_{\tau \in [0,T]}$. In other words, this stochastic process induces a joint distribution of $(\boldsymbol{X}_{\tau_1}, \ldots, \boldsymbol{X}_{\tau_n})$. Our goal is to model $\{\boldsymbol{X}_\tau\}_{\tau \in [0,T]}$ such that the log-likelihood of the observations

$$\mathcal{L} = \log p_{\boldsymbol{X}_{\tau_1}, \ldots, \boldsymbol{X}_{\tau_n}}(\boldsymbol{x}_{\tau_1}, \ldots, \boldsymbol{x}_{\tau_n}) \tag{5}$$

is maximized. We define the continuous-time flow process (CTFP) $\{F_{\boldsymbol{\theta}}(\boldsymbol{W}_\tau; \tau)\}_{\tau \in [0,T]}$ such that

$$\boldsymbol{X}_\tau = F_{\boldsymbol{\theta}}(\boldsymbol{W}_\tau; \tau), \quad \forall \tau \in [0,T], \tag{6}$$

where $F_{\boldsymbol{\theta}}(\cdot; \tau) : \mathbb{R}^d \to \mathbb{R}^d$ is an invertible mapping parametrized by the learnable parameters $\boldsymbol{\theta}$ for every $\tau \in [0,T]$, and $\boldsymbol{W}_\tau$ is a $d$-dimensional Wiener process.

The log-likelihood in Equation 5 can be rewritten using the change of variables formula. Let $\boldsymbol{w}_{\tau_i} = F_{\boldsymbol{\theta}}^{-1}(\boldsymbol{x}_{\tau_i}; \tau_i)$, then

$$\mathcal{L} = \sum_{i=1}^n \left[ \log p_{\boldsymbol{W}_{\tau_i} | \boldsymbol{W}_{\tau_{i-1}}}(\boldsymbol{w}_{\tau_i} | \boldsymbol{w}_{\tau_{i-1}}) - \log \left| \det \frac{\partial F_{\boldsymbol{\theta}}(\boldsymbol{w}_{\tau_i}; \tau_i)}{\partial \boldsymbol{w}_{\tau_i}} \right| \right], \tag{7}$$

where $\tau_0 = 0$, $\boldsymbol{W}_0 = 0$, and $p_{\boldsymbol{W}_{\tau_i}|\boldsymbol{W}_{\tau_{i-1}}}$ is defined in Section 3.1. Figure 2a shows an example of the likelihood calculation. Sampling from a CTFP is straightforward: given the timestamps $\tau_i$, we first sample a realization of the Wiener process $\{\boldsymbol{w}_{\tau_i}\}_{i=1}^n$, then map them to $\boldsymbol{x}_{\tau_i} = F_{\boldsymbol{\theta}}(\boldsymbol{w}_{\tau_i}; \tau_i)$. Figure 2b illustrates this procedure.

The normalizing flow $F_{\boldsymbol{\theta}}(\cdot; \tau)$ transforms a simple base distribution induced by $\boldsymbol{W}_\tau$ on an arbitrary time grid into a more complex shape in the observation space. It is worth noting that given a continuous realization of $\boldsymbol{W}_\tau$, as long as $F_{\boldsymbol{\theta}}(\cdot; \tau)$ is implemented as a continuous mapping, the resulting trajectory $\boldsymbol{x}_\tau$ is also continuous.

## 4.2 Generative ANODE

In principle, any normalizing flow model indexed by time $\tau$ could be used as $F_{\boldsymbol{\theta}}(\cdot; \tau)$ in Equation 6. We proceed with the continuous normalizing flow and ANODE, because it has a free-form Jacobian and efficient trace estimator [15, 21]. In particular, we consider the following instantiation of ANODE as a generative model: For any $\tau \in [0, T]$ and $\boldsymbol{w}_\tau \in \mathbb{R}^d$, we map $\boldsymbol{w}_\tau$ to $\boldsymbol{x}_\tau$ by solving the following initial value problem:

$$\frac{d}{dt}\begin{pmatrix} \boldsymbol{h}_\tau(t) \\ a_\tau(t) \end{pmatrix} = \begin{pmatrix} f_{\boldsymbol{\theta}}(\boldsymbol{h}_\tau(t), a_\tau(t), t) \\ g_{\boldsymbol{\theta}}(a_\tau(t), t) \end{pmatrix}, \quad \begin{pmatrix} \boldsymbol{h}_\tau(t_0) \\ a_\tau(t_0) \end{pmatrix} = \begin{pmatrix} \boldsymbol{w}_\tau \\ \tau \end{pmatrix}, \quad (8)$$

where $\boldsymbol{h}_\tau(t) \in \mathbb{R}^d, t \in [t_0, t_1]$, $f_{\boldsymbol{\theta}} : \mathbb{R}^d \times \mathbb{R} \times [t_0, t_1] \to \mathbb{R}^d$, and $g_{\boldsymbol{\theta}} : \mathbb{R} \times [t_0, t_1] \to \mathbb{R}$. Then $F_{\boldsymbol{\theta}}$ in Equation 6 is defined as the solution of $\boldsymbol{h}_\tau(t)$ at $t = t_1$:

$$F_{\boldsymbol{\theta}}(\boldsymbol{w}_\tau; \tau) := \boldsymbol{h}_\tau(t_1) = \boldsymbol{h}_\tau(t_0) + \int_{t_0}^{t_1} f_{\boldsymbol{\theta}}(\boldsymbol{h}_\tau(t), a_\tau(t), t) \, dt. \quad (9)$$

Note that the index $t$ represents the independent variable in the initial value problem and should not be confused with $\tau$, the timestamp of the observation.

Using Equation 4, the log-likelihood $\mathcal{L}$ can be calculated as follows:

$$\mathcal{L} = \sum_{i=1}^n \left[ \log p_{\boldsymbol{W}_{\tau_i}|\boldsymbol{W}_{\tau_{i-1}}}\left(\boldsymbol{h}_{\tau_i}(t_0)|\boldsymbol{h}_{\tau_{i-1}}(t_0)\right) - \int_{t_0}^{t_1} \text{tr}\left(\frac{\partial f_{\boldsymbol{\theta}}(\boldsymbol{h}_{\tau_i}(t), a_{\tau_i}(t), t)}{\partial \boldsymbol{h}_{\tau_i}(t)}\right) dt \right], \quad (10)$$

where $\boldsymbol{h}_{\tau_i}(t_0)$ is obtained by solving the ODE in Equation 8 backwards from $t = t_1$ to $t = t_0$, and the trace of the Jacobian can be estimated by Hutchinson's trace estimator [22, 21].

## 4.3 Interpolation and Extrapolation with CTFP

Time-indexed normalizing flows and the Brownian bridge allow us to define conditional distributions on arbitrary timestamps. They also permit the CTFP model to perform interpolation and extrapolation given partial observations, which are of great importance in time series modeling.

Interpolation means that we can model the conditional distribution $p_{\boldsymbol{X}_\tau|\boldsymbol{X}_{\tau_i},\boldsymbol{X}_{\tau_{i+1}}}(\boldsymbol{x}_\tau|\boldsymbol{x}_{\tau_i}, \boldsymbol{x}_{\tau_{i+1}})$ for all $\tau \in [\tau_i, \tau_{i+1}]$ and $i = 1, \ldots, n-1$. This can be done by mapping the values $\boldsymbol{x}_\tau, \boldsymbol{x}_{\tau_i}$ and $\boldsymbol{x}_{\tau_{i+1}}$ to $\boldsymbol{w}_\tau, \boldsymbol{w}_{\tau_i}$ and $\boldsymbol{w}_{\tau_{i+1}}$, respectively. After that, Equation 2 can be applied to obtain the conditional density of $p_{\boldsymbol{W}_\tau|\boldsymbol{W}_{\tau_i},\boldsymbol{W}_{\tau_{i+1}}}(\boldsymbol{w}_\tau|\boldsymbol{w}_{\tau_i}, \boldsymbol{w}_{\tau_{i+1}})$. Finally, we have

$$\log p_{\boldsymbol{X}_\tau|\boldsymbol{X}_{\tau_i},\boldsymbol{X}_{\tau_{i+1}}}(\boldsymbol{x}_\tau|\boldsymbol{x}_{\tau_i}, \boldsymbol{x}_{\tau_{i+1}}) = \log p_{\boldsymbol{W}_\tau|\boldsymbol{W}_{\tau_i},\boldsymbol{W}_{\tau_{i+1}}}(\boldsymbol{w}_\tau|\boldsymbol{w}_{\tau_i}, \boldsymbol{w}_{\tau_{i+1}}) - \log\left|\det\frac{\partial \boldsymbol{x}_\tau}{\partial \boldsymbol{w}_\tau}\right|. \quad (11)$$

Extrapolation can be done in a similar fashion using Equation 1. This allows the model to predict continuous trajectories into the future, given past observations. Figure 2c shows a visualization of interpolation and extrapolation using CTFP.

## 4.4 Latent Continuous-Time Flow Process

The CTFP model inherits the Markov property from the Wiener process, which is a strong assumption and limits its ability to model stochastic processes with complex temporal dependencies. In order to enhance the expressive power of the CTFP model, we augment it with a latent variable $\boldsymbol{Z} \in \mathbb{R}^m$,

whose prior distribution is an isotropic Gaussian $p_{\boldsymbol{Z}}(\boldsymbol{z}) = \mathcal{N}(\boldsymbol{z}; 0, \boldsymbol{I}_m)$. As a result, the data distribution can be approximated by a diverse collection of CTFP models conditioned on sampled latent variables $\boldsymbol{z}$.

The generative model in Equation 6 is augmented to $\boldsymbol{X}_\tau = F_{\boldsymbol{\theta}}(\boldsymbol{W}_\tau; \boldsymbol{Z}, \tau), \forall \tau \in [0, T]$, which induces the conditional distribution $\boldsymbol{X}_{\tau_1}, \ldots, \boldsymbol{X}_{\tau_n} | \boldsymbol{Z}$. Similar to the initial value problem in Equation 8, we define $F_{\boldsymbol{\theta}}(\boldsymbol{w}_\tau; \boldsymbol{z}, \tau) = \boldsymbol{h}_\tau(t_1)$, where

$$\frac{d}{dt}\begin{pmatrix} \boldsymbol{h}_\tau(t) \\ \boldsymbol{a}_\tau(t) \end{pmatrix} = \begin{pmatrix} f_{\boldsymbol{\theta}}(\boldsymbol{h}_\tau(t), \boldsymbol{a}_\tau(t), t) \\ g_{\boldsymbol{\theta}}(\boldsymbol{a}_\tau(t), t) \end{pmatrix}, \begin{pmatrix} \boldsymbol{h}_\tau(t_0) \\ \boldsymbol{a}_\tau(t_0) \end{pmatrix} = \begin{pmatrix} \boldsymbol{w}_\tau \\ (\boldsymbol{z}, \tau)^\top \end{pmatrix}. \tag{12}$$

Depending on the sample of the latent variable $\boldsymbol{z}$, the CTFP model has different gradient fields and thus different output distributions.

For simplicity of notation, the subscripts of density functions are omitted from now on. For the augmented generative model, the log-likelihood becomes $\mathcal{L} = \log \int_{\mathbb{R}^m} p(\boldsymbol{x}_{\tau_1}, \ldots, \boldsymbol{x}_{\tau_n} | \boldsymbol{z}) p(\boldsymbol{z}) \, d\boldsymbol{z}$, which is intractable to evaluate. Following the variational autoencoder approach [28], we introduce an approximate posterior distribution of $\boldsymbol{Z} | \boldsymbol{X}_{\tau_1}, \ldots, \boldsymbol{X}_{\tau_n}$, denoted by $q(\boldsymbol{z} | \boldsymbol{x}_{\tau_1}, \ldots, \boldsymbol{x}_{\tau_n})$. The implementation of the approximate posterior distribution is an ODE-RNN encoder [42]. With the approximate posterior distribution, we can derive an importance-weighted autoencoder (IWAE) [6] lower bound of the log-likelihood on the right-hand side of the inequality:

$$\begin{aligned} \mathcal{L} &= \log \mathbb{E}_{\boldsymbol{z} \sim q} \left[ \frac{p(\boldsymbol{x}_{\tau_1}, \ldots, \boldsymbol{x}_{\tau_n} | \boldsymbol{z}) p(\boldsymbol{z})}{q(\boldsymbol{z} | \boldsymbol{x}_{\tau_1}, \ldots, \boldsymbol{x}_{\tau_n})} \right] \\ &\geq \mathbb{E}_{\boldsymbol{z}_1, \ldots, \boldsymbol{z}_K \sim q} \left[ \log \left( \frac{1}{K} \sum_{k=1}^{K} \frac{p(\boldsymbol{x}_{\tau_1}, \ldots, \boldsymbol{x}_{\tau_n} | \boldsymbol{z}_k) p(\boldsymbol{z}_k)}{q(\boldsymbol{z}_k | \boldsymbol{x}_{\tau_1}, \ldots, \boldsymbol{x}_{\tau_n})} \right) \right] =: \mathcal{L}_{\text{IWAE}}, \end{aligned} \tag{13}$$

where $K$ is the number of samples from the approximate posterior distribution.

## 5 Experiments

In this section, we apply our models on synthetic data generated from common continuous-time stochastic processes and complex real-world datasets. The proposed CTFP and latent CTFP models are compared against two baseline models: latent ODEs [42] and variational RNNs (VRNNs) [12]. The latent ODE model with the ODE-RNN encoder is designed specifically to model time series data with irregular observation times. VRNN is a popular variational filtering model that demonstrated superior performance on structured sequential data.

For VRNNs, we append the time gap between two observations as an additional input to the neural network. Both latent CTFP and latent ODE models use ODE-RNN [42] as the inference network; GRU [11] is used as the RNN cell in latent CTFP, latent ODE, and VRNN models. All three latent variable models have the same latent dimension and GRU hidden state dimension. Please see the supplementary materials for details about our experimental setup and model implementations.

### 5.1 Synthetic Datasets

We simulate three irregularly-sampled time series datasets; all of them are univariate. **Geometric Brownian motion** (GBM) is a continuous-time stochastic process widely used in mathematical finance. It satisfies the following stochastic differential equation: $dX_\tau = \mu X_\tau d\tau + \sigma X_\tau dW_\tau$, where $\mu$ and $\sigma$ are the drift term and variance term, respectively. The timestamps of the observations are in the range between 0 and $T = 30$ and are sampled from a homogeneous Poisson point process with an intensity of $\lambda_{\text{train}} = 2$. To further evaluate the model's capacity to capture the dynamics of GBM, we test the model with observation time-steps sampled from Poisson point processes with intensities of $\lambda_{\text{test}} = 2$ and $\lambda_{\text{test}} = 20$. **Ornstein–Uhlenbeck process** (OU Process) is another type of widely used continuous-time stochastic process. The OU process satisfies the following stochastic differential equation: $dX_\tau = \theta(\mu - X_\tau)d\tau + \sigma dW_\tau$. We use the same set of observation intensities as in our GBM experiments to sample observation timestamps in the training and test sets. **Mixture of OUs.** To demonstrate the latent CTFP's capability to model sequences sampled from different continuous-time stochastic processes, we train our models on a dataset generated by mixing the sequences sampled from two different OU processes with different values of $\theta, \mu, \sigma$, and

Table 1: **Quantitative Evaluation (Synthetic Data)**. We show test negative log-likelihood on three synthetic stochastic processes across different models. Below each process, we indicate the intensity of the Poisson point process from which the timestamps for the test sequences were sampled. "Ground Truth" refers to the closed-form negative log-likelihood of the true underlying data generation process. [GBM: geometric Brownian motion; OU: Ornstein–Uhlenbeck process; M-OU: mixture of OUs.]

| Model | GBM | | OU | | M-OU |
|---|---|---|---|---|---|
| | $\lambda_{test} = 2$ | $\lambda_{test} = 20$ | $\lambda_{test} = 2$ | $\lambda_{test} = 20$ | $\lambda_{test} = (2, 20)$ |
| Latent ODE [42] | 3.826 | 5.935 | 3.066 | 3.027 | 2.690 |
| VRNN [12] | 3.762 | 3.492 | **2.729** | **1.939** | 1.415 |
| CTFP (**ours**) | **3.107** | **1.929** | 2.902 | 1.941 | 1.408 |
| Latent CTFP (**ours**) | **3.107** | 1.930 | 2.902 | **1.939** | **1.392** |
| Ground Truth | 3.106 | 1.928 | 2.722 | 1.888 | 1.379 |

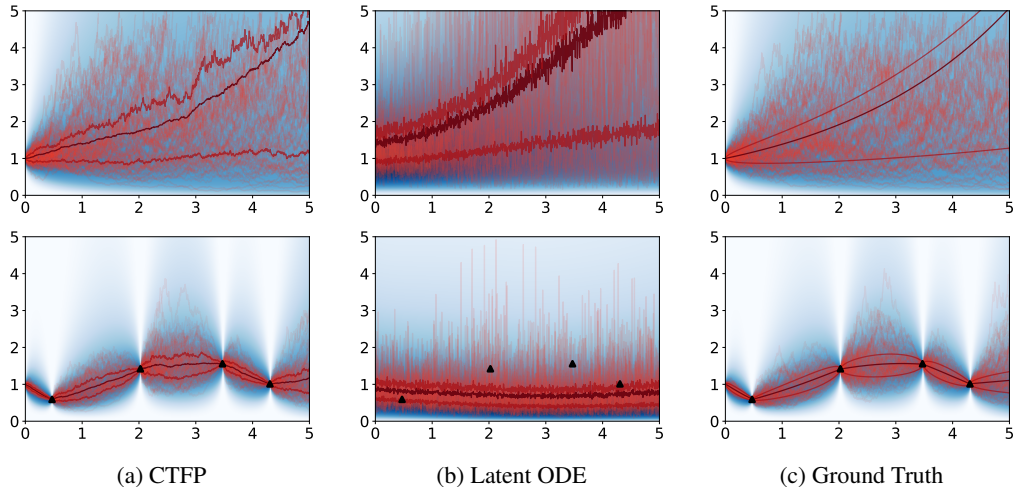

(a) CTFP            (b) Latent ODE            (c) Ground Truth

Figure 3: **Comparison between CTFP and latent ODE on the GBM data.** We consider the generation and interpolation tasks for (a) CTFP, (b) latent ODE, and (c) ground truth. In each subfigure, the upper panel shows samples generated from the model and the lower panel shows results for interpolation. The observed points for interpolation are marked by black triangles. In addition to the sample trajectories (red) and the marginal density (blue), we also show the sample-based estimates (closed-form for ground truth) of the inter-quartile range (dark red) and mean (brown) of the marginal density.

different observation intensities. We defer the details of the parameters of the synthetic dataset to the supplementary materials.

**Results.** The results are presented in Table 1. We report the exact negative log-likelihood (NLL) per observation for CTFP. For latent ODE, latent CTFP, and VRNN, we report the (upper bound of) NLL estimated by the IWAE bound [6] in Equation 13, using $K = 25$ samples of latent variables. We also show the NLL of the test set computed with the ground truth density function.

The results on the test set sampled from the GBM indicate that the CTFP model can recover the true data generation process as the NLL estimated by CTFP is close to the ground truth. In contrast, latent ODE and VRNN models fail to recover the true data distribution. On the M-OU dataset, the latent CTFP models show better performance than the other models. Moreover, latent CTFP outperforms CTFP by 0.016 nats, indicating its ability to leverage the latent variables.

Although trained on samples with an observation intensity of $\lambda_{train} = 2$, CTFP can better adapt to samples with a bigger observation intensity (and thus denser time grid) of $\lambda_{test} = 20$. We hypothesize that the superior performance of CTFP models when $\lambda_{test} = 20$ is due to their capability to model continuous stochastic processes, whereas the baseline models do not have the notion of continuity. We further corroborate this hypothesis in an ablation study where the base Wiener process is replaced with i.i.d. Gaussian random variables, such that the base process is no longer continuous in time (see the supplementary materials).

Table 2: **Quantitative Evaluation (Real-World Data).** We show test negative log-likelihood on Mujoco-Hopper, Beijing Air-Quality Dataset (BAQD) and PTB Diagnostic Database (PTBDB) across different models. For CTFP, the reported values are exact; for the other three models, we report IWAE bounds using $K = 125$ samples. Lower values correspond to better performance. Standard deviations are based on 5 independent runs.

| Model | Mujoco-Hopper [42] | BAQD [4] | PTBDB [47] |
|---|---|---|---|
| Latent ODE [42] | $24.775 \pm 0.010$ | $2.789 \pm 0.011$ | $-0.818 \pm 0.009$ |
| VRNN [12] | $9.113 \pm 0.018$ | $0.604 \pm 0.007$ | $\mathbf{-1.999 \pm 0.008}$ |
| CTFP (**ours**) | $-16.249 \pm 0.034$ | $-2.361 \pm 0.020$ | $-1.324 \pm 0.028$ |
| Latent CTFP (**ours**) | $\mathbf{-31.397 \pm 0.063}$ | $\mathbf{-6.894 \pm 0.046}$ | $-1.999 \pm 0.010$ |

Figure 3 provides a qualitative comparison between CTFP and latent ODE trained on the GBM data, both on the generation task (upper panels) and the interpolation task (lower panels). The results in the upper panels show that CTFP can generate continuous sample paths and accurately estimate the marginal mean and quantiles. In contrast, the sample paths generated by latent ODE are more volatile and discontinuous due to its lack of continuity. For the interpolation task, the results of CTFP are consistent with the ground truth in terms of both point estimation and uncertainty estimation. For latent ODE on the interpolation task, Figure 3b shows that the latent variables from the variational posterior shift the density to the region where the observations lie. However, although latent ODE is capable of performing interpolation, there is no guarantee that the (reconstructed) sample paths pass through the observed points (triangular marks in Figure 3b), as discussed in Section 2. In addition to these difficulties with the interpolation task, the qualitative comparison between samples further highlights the importance of our models' continuity when generating samples of continuous dynamics.

## 5.2 Real-World Datasets

We also evaluate our models on real-world datasets with continuous and complex dynamics. The following three datasets are considered: **Mujoco-Hopper** [42] consists of 10,000 sequences that are simulated by a "Hopper" model from the DeepMind Control Suite in a MuJoCo environment [46]. **PTB Diagnostic Database** (PTBDB) [4] consists of excerpts of ambulatory electrocardiography (ECG) recordings. Each sequence is one-dimensional and the sampling frequency of the recordings is 125 Hz. **Beijing Air-Quality Dataset** (BAQD) [47] is a dataset consisting of multi-year recordings of weather and air quality data across different locations in Beijing. The variables in consideration are temperature, pressure, and wind speed, and the values are recorded once per hour. We segment the data into sequences, each covering the recordings of a whole week. Please refer to the supplementary materials for additional details about data preprocessing.

Similar to our synthetic data experiment settings, we compare the CTFP and latent CTFP models against latent ODE and VRNN. It is worth noting that the latent ODE model in the original work [42] uses a fixed output variance and is evaluated using mean squared error (MSE); we adapt the model to our tasks with a predicted output variance (see supplementary materials). We further study the effect of using RealNVP [14] as the invertible mapping $F_{\boldsymbol{\theta}}(\cdot; \tau)$. This experiment can be regarded as an ablation study and results are presented in the supplementary materials as well.

**Results.** The results are shown in Table 2. We report the exact negative log-likelihood (NLL) per observation for CTFP and the (upper bound of) NLL estimated by the IWAE bound, using $K = 125$ samples of latent variables, for latent ODE, latent CTFP, and VRNN. For each setting, the mean and standard deviation of five evaluation runs are reported. The evaluation results show that the latent CTFP model outperforms VRNN and latent ODE models on real-world datasets, indicating that CTFP is better at modeling irregular time series data with continuous dynamics. Table 2 also suggests that the latent CTFP model consistently outperforms the CTFP model, demonstrating that with the latent variables, the latent CTFP model is more expressive and able to capture the data distribution better. We defer additional experimental results with different observation processes but similar conclusions to the supplementary materials.

# 6 Conclusion

In summary, we propose the continuous-time flow process (CTFP), a reversible generative model for stochastic processes, and its latent variant. It maps a simple continuous-time stochastic process, i.e., the Wiener process, into a more complicated process in the observable space. As a result, many desirable mathematical properties of the Wiener process are retained, including the efficient sampling of continuous paths, likelihood evaluation on arbitrary timestamps, and inter-/extrapolation given observed data. Our experimental results demonstrate the superior performance of the proposed models on various datasets.

## Broader Impact Statement

Time series models could be applied to a wide range of applications, including natural language processing, recommendation systems, traffic prediction, medical data analysis, forecasting, and others. Our research improves over the existing models on a particular type of data: irregular time series data.

There are opportunities for applications using the proposed models for beneficial purposes, such as weather forecasting, pedestrian behavior prediction for self-driving cars, and missing healthcare data interpolation or prediction. We encourage practitioners to understand the impacts of using CTFP in particular real-world scenarios.

One potential risk is that the capability of interpolation and extrapolation can be used in malicious ways. An adversary might be able to use the proposed model to infer private information given partial observations, which leads to privacy concerns. We would encourage further research to address this risk using tools like differential privacy.

## Funding Transparency Statement

This work was conducted at Borealis AI and partly supported by Mitacs through the Mitacs Accelerate program.

## Footnotes

*Work developed during an internship at Borealis AI. Correspond to wsdmdeng@gmail.com.

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
