[Supplementary Material]

# Modeling Continuous Stochastic Processes with Dynamic Normalizing Flows Supplementary Materials

**Ruizhi Deng**[1,2*]  **Bo Chang**[1]  **Marcus A. Brubaker**[1,3,4]  **Greg Mori**[1,2]  **Andreas Lehrmann**[1]
[1]Borealis AI   [2]Simon Fraser University   [3]York University   [4]Vector Institute

## A  Finite-Dimensional Distribution of CTFP

Equation 7 in Section 4 is the log density of the distribution obtained by applying the normalizing flow models to the finite-dimensional distribution of Wiener process on a given time grid. A natural question that would arise is why the distribution described by Equation 7 necessarily matches the finite-dimensional distribution of $\boldsymbol{X}_\tau = F_{\boldsymbol{\theta}}(\boldsymbol{W}_\tau, \tau)$. In other words, it is left to close the gap between the distributions of samples obtained by two different ways to justify Equation 7: (1) first getting a sample path of $\boldsymbol{X}_\tau$ by applying the transformation defined by $F_{\boldsymbol{\theta}}$ to a sample of $\boldsymbol{W}_\tau$ and then obtaining the finite-dimensional observation of $\boldsymbol{X}_\tau$ on the time grid; (2) first obtaining the finite-dimensional sample of $\boldsymbol{W}_\tau$ and applying the normalizing flows to this finite-dimensional distribution. To justify the finite-dimensional distribution of CTFP, we choose to work with the canonical Wiener space $(\Omega, \Sigma)$ equipped with the unique Wiener measure $\boldsymbol{\mu_W}$ where $\Omega = C([0, +\infty), \mathbb{R}^d)$ is the set of continuous functions from $[0, +\infty)$ to $\mathbb{R}^d$, $\Sigma$ is the Borel $\sigma$-algebra generated by all the cylinder sets of $C([0, +\infty), \mathbb{R}^d)$, and $\boldsymbol{W}_\tau(\boldsymbol{\omega}) = \boldsymbol{\omega}(\tau)$ for $\boldsymbol{\omega} \in \Omega$. We refer the reader to Chapter 2 of [5] for more details. Given a time grid $0 < \tau_1 < \tau_2 < \cdots < \tau_n$, the distribution of observations of Wiener process on this discrete time grid is called the finite-dimensional distribution of $\boldsymbol{W}_\tau$. It is a push-forward measure on $(\mathbb{R}^{d \times n}, \mathcal{B}(\mathbb{R}^{d \times n}))$ induced by the projection mapping $\pi_{\tau_1, \tau_2, \ldots, \tau_n} : (\Omega, \Sigma) \to ((\mathbb{R}^{d \times n}, \mathcal{B}(\mathbb{R}^{d \times n})))$ on this grid where $\mathcal{B}(\cdot)$ denotes the Borel $\sigma$-algebra. Therefore, for each Borel (measurable) set $B$ of $\mathbb{R}^{d \times n}$, the finite-dimensional distribution of $B$ is $\boldsymbol{\mu_W} \circ \pi^{-1}(B) = \boldsymbol{\mu_W}(\{\boldsymbol{\omega} | (\boldsymbol{W}_{\tau_1}(\boldsymbol{\omega}) \ldots \boldsymbol{W}_{\tau_n}(\boldsymbol{\omega})) \in B\})$. We drop the subscript of $\pi$ for the simplicity of notation. We base the justification on the following two propositions.

**Proposition 1.** *Let $F_{\boldsymbol{\theta}}(\cdot, \cdot)$ be defined as Equation 8 and 9 in Section 4.2. The mapping from $(\Omega, \Sigma, \boldsymbol{\mu_W})$ to $(\Omega, \Sigma)$ defined by $\boldsymbol{\omega}(\tau) \to F_{\boldsymbol{\theta}}(\boldsymbol{\omega}(\tau), \tau)$ is measurable and therefore induces a pushforward measure $\boldsymbol{\mu_W} \circ F_{\boldsymbol{\theta}}^{-1}$.*

*Proof.* As $F_\theta$ is continuous in both $\boldsymbol{\omega}$ and $\tau$, it is easy to show $F_\theta(\boldsymbol{\omega}(\tau), \tau)$ is also continuous in $\tau$ for each $\omega$ continuous in $\tau$. As $F_\theta(\cdot, \tau)$ is invertible for each $\tau$, $F_\theta(\cdot, \tau)$ is an homeomorphsim between $\mathbb{R}^d$ and $\mathbb{R}^d$. Therefore, the pre-image of each Borel set of $\mathbb{R}^d$ under $F_\theta(\cdot, \tau)$ for each $\tau$ is also Borel. As a result, the pre-image of each cylinder set of $C([0, +\infty), \mathbb{R}^d)$ under the mapping defined by $F_\theta(\cdot, \cdot)$ is also a cylinder set, which is enough to show the mapping is measurable.  $\square$

This proposition shows $\boldsymbol{X}_\tau$ is a stochastic process also defined in the space of continuous functions as Wiener process. It provides a solid basis to for defining finite-dimensional distribution of $\boldsymbol{X}_\tau$ on $\mathbb{R}^{d \times n}$ in a similar ways as Wiener process using projection. The two sampling methods mentioned above can be characterized by two different mappings from $(\Omega, \Sigma, \boldsymbol{\mu_W})$ to $(\mathbb{R}^{d \times n}, \mathcal{B}(\mathbb{R}^{d \times n}))$: (1) applying transformation defined by $F_\theta$ to a function in $C([0, +\infty), \mathbb{R}^d)$ and then applying the projection $\pi$ to the transformed function given a time grid; (2) applying the projection to a continuous function on a time grid and applying the transformation defined by $F_\theta(\cdot, \tau)$ for each $\tau$ individually. We can check

the pushforward measures induced by the two mappings agree on every Borel set of $\mathbb{R}^{d \times n}$ as their pre-images are the same in $(\Omega, \Sigma, \boldsymbol{\mu_W})$. Therefore we have the following proposition:

**Proposition 2.** *Given a finite subset $\{\tau_1, \tau_2, ..., \tau_n\} \subset (0, +\infty)$, the finite-dimensional distribution of $\boldsymbol{X}_\tau$ is the same as the distribution of $(F_\theta(\boldsymbol{W}_{\tau_1}, \tau_1), ..., F_\theta(\boldsymbol{W}_{\tau_n}, \tau_n))$, where $(\boldsymbol{W}_{\tau_1}, ..., \boldsymbol{W}_{\tau_n})$ is a $n \times d$-dimensional random variable with finite-dimensional distribution of $\boldsymbol{W}_\tau$.*

*Proof.* It suffices to check that given the fixed time grid, for each Borel set $B \subset \mathbb{R}^{d \times n}$, the preimage of $B$ is the same under the two mappings. They are both $\{\boldsymbol{\omega} | (F_\theta(\boldsymbol{W}_{\tau_1}(\boldsymbol{\omega}), \tau_1), F_\theta(\boldsymbol{W}_{\tau_2}(\boldsymbol{\omega}), \tau_2), \dots, F_\theta(\boldsymbol{W}_{\tau_n}(\boldsymbol{\omega}), \tau_n)) \in B\}$. $\qquad\square$

# B  Experiment Setup and Model Architecture Details

We describe the details on synthetic dataset generation, real-world dataset pre-processing, model architecture as well as training and evaluation settings in this section.

## B.1  Synthetic Dataset Details

For the geometric Brownian motion (GBM), we sample 10000 trajectories from a GBM with the parameters of $\mu = 0.2$ and a variance of $\sigma = 0.5$ in the interval of [0, 30]. The timestamps of the observations are sampled from a homogeneous Poisson point process with an intensity of $\lambda_{\text{train}} = 2$. We evaluate the model on the observations timestamps sampled from two homogeneous Poisson processes separately with intensity values of $\lambda_{\text{test}} = 2$ and $\lambda_{\text{test}} = 20$.

For the Ornstein–Uhlenbeck (OU) process, the parameters of the process we sample trajectories from are $\theta = 2, \mu = 1$, and $\sigma = 10$. We also sample 10000 trajectories and use the same set of observation intensity values, $\lambda_{\text{train}}$ and $\lambda_{\text{test}}$, to sample observation timestamps from homogeneous Poisson processes for training and test.

For the mixture of OU processes (MOU), we sample 5000 sequences from each of two different OU processes and mix them to obtain 10000 sequences. One OU process has the parameters of $\theta = 2, \mu = 1$, and $\sigma = 10$ and the observation timestamps are sampled from a homogeneous Poisson process with $\lambda_{\text{train}} = 2$. The other OU process has the parameters of $\theta = 1.0, \mu = 2.0$, and $\sigma = 5.0$ with observation timestamps sampled with $\lambda_{\text{train}} = 20$.

For the 10000 trajectories of each dataset, we use 7000 trajectories for training and 1000 trajectories for validation. We test the model on 2000 trajectories for each value of $\lambda_{\text{test}}$. To test the model with $\lambda_{\text{test}} = 20$ on GBM and OU process, we also use 2000 sequences.

## B.2  Real-World Dataset Details

As mentioned in Section 5.2 of the paper, we compare our models against the baselines on three datasets: Mujoco-Hopper, Beijing Air-Quality dataset (BAQD), and PTB Diagnostic Database(PTBDB). The three datasets can be downloaded using the following links:

- `http://www.cs.toronto.edu/~rtqichen/datasets/HopperPhysics/training.pt`
- `https://www.kaggle.com/shayanfazeli/heartbeat/download`
- `https://archive.ics.uci.edu/ml/datasets/Beijing+Multi-Site+Air-Quality+Data`

We pad all sequences into the same length for each dataset. The sequence length of the Mujoco-Hopper dataset is 200 and the sequence length of BAQD is 168. The maximum sequence length in the PTBDB dataset is 650. We rescale the indices of sequences to real numbers in the interval of [0, 120] and take the rescaled values as observation timestamps for all datasets. To make the sequences asynchronous or irregularly-sampled, we sample observation timestamps $\{\tau_i\}_{i=1}^n$ from a homogeneous Poisson process with an intensity of 2 that is independent of the data. For each sampled timestamp, the value of the closest observation is taken as its corresponding value. The timestamps of all sampled sequences are shifted by a value of 0.2 since $\boldsymbol{W}_0 = 0$ deterministically for the Wiener process and there's no variance for the CTFP model's prediction at $\tau = 0$.

## B.3 Model Architecture Details

To ensure a fair comparison, we use the same values for hyper-parameters including the latent variable and hidden state dimensions across all models. Likewise, we keep the underlying architectures as similar as possible and use the same experimental protocol across all models.

For CTFP and Latent CTFP, we use a one-block augmented neural ODE module that maps the base process to the observation process. For the augmented neural ODE model, we use an MLP model consisting of 4 hidden layers of size 32–64–64–32 for the model in Equation 8 and Equation 12. In practice, the implementation of $g$ in the two equations is optional and its representation power can be fully incorporated into $f$. This architecture is used for both synthetic and real-world datasets. For the latent CTFP and latent ODE models appearing in Section 5, we use the ODE-RNN model as the recognition network. For synthetic datasets, the ODE-RNN model consists of a one-layer GRU cell with a hidden dimension of 20 (the rec-dims parameter in its original implementation) and a one-block neural ODE module that has a single hidden layer of size 100, and it outputs a 10-dimensional latent variable. The same architecture is used by both latent ODE and latent CTFP models. For real-world datasets, the ODE-RNN architecture uses a hidden state of dimension 20 in the GRU cell and an MLP with a 128-dimensional hidden layer in the neural ODE module. The ODE-RNN model produces a 64-dimensional latent variable. For the generation network of the latent ODE (V2) model, we use an ODE function with one hidden layer of size 100 for synthetic datasets and 128 for real-world datasets. The decoder network has 4 hidden layers of size 32–64–64–32; it maps a latent trajectory to outputs of Gaussian distributions at different time steps.

The VRNN model is implemented using a GRU network. The hidden state of the VRNN models is 20-dimensional for synthetic and real-world datasets. The dimension of the latent variable is 64 for real-word datasets and 10 for synthetic datasets. We use an MLP of 4 hidden layers of size 32–64–64–32 for the decoder network, an MLP with one hidden layer that has the same dimension as the hidden state for the prior proposal network, and an MLP with two hidden layers for the posterior proposal network. For synthetic data sampled from Geometric Brownian Motion, we apply an exponential function to the samples of all models. Therefore the distribution precited by latent ODE and VRNN at each timestamp is a log-normal distribution.

## B.4 Training and Evaluation Settings

For synthetic data, we train all models using the IWAE bound with 3 samples and a flat learning rate of $5 \times 10^{-4}$ for all models. We also consider models trained with or without the aggressive training scheme proposed by He et al. [4] for latent ODE and latent CTFP. We choose the best-performing model among the ones trained with or without the aggressive scheme based IWAE bound, estimated with 25 samples on the validation set for evaluation. The batch size is 100 for CTFP models and 25 for all the other models. For experiments on real-world datasets, we did a hyper-parameter search on learning rates over two values of $5 \times 10^{-4}$ and $10^{-4}$, and whether using the aggressive training schemes for latent CTFP and latent ODE models. We report the evaluation results of the best-performing model based on IWAE bound estimated with 125 samples.

# C Ablation Study Results

## C.1 Additional Experiment Results on Real-world Datasets

We provide additional experiment results on real-world datasets using different intensity value $\lambda$s of 1 and 5 to sample observation processes in Table 1 below.

Table 1: Ablation Study on Time Interval for Real-World Data

| Model | Negative Log-Likelihood | | | | | |
| --- | --- | --- | --- | --- | --- | --- |
| | Mujoco-Hopper | | BAQD | | PTBDB | |
| | $\lambda_{\text{test}} = 1$ | $\lambda_{\text{test}} = 5$ | $\lambda_{\text{test}} = 1$ | $\lambda_{\text{test}} = 5$ | $\lambda_{\text{test}} = 1$ | $\lambda_{\text{test}} = 5$ |
| Latent ODE | $25.082 \pm 0.011$ | $24.599 \pm 0.004$ | $2.948 \pm 0.006$ | $2.686 \pm 0.006$ | $-0.633 \pm 0.006$ | $-0.892 \pm 0.009$ |
| VRNN | $10.553 \pm 0.010$ | $8.543 \pm 0.008$ | $0.044 \pm 0.007$ | $-1.016 \pm 0.001$ | $\mathbf{-1.552 \pm 0.011}$ | $\mathbf{-2.545 \pm 0.005}$ |
| CTFP | $-10.152 \pm 0.084$ | $-23.241 \pm 0.057$ | $-1.255 \pm 0.022$ | $-3.784 \pm 0.035$ | $-1.028 \pm 0.028$ | $-1.824 \pm 0.014$ |
| Latent CTFP | $\mathbf{-30.469 \pm 0.079}$ | $\mathbf{-33.412 \pm 0.035}$ | $\mathbf{-7.276 \pm 0.061}$ | $\mathbf{-6.226 \pm 0.016}$ | $-1.552 \pm 0.010$ | $-2.533 \pm 0.008$ |

Table 2: Comparison between CTFP, CTFP-IID-Gaussian, latent CTFP, and latent CTFP-IID-Gaussian on synthetic datasets. We report NLL per observation.

| Model | GBM | | OU | | M-OU |
|---|---|---|---|---|---|
| | $\lambda_{\text{test}} = 2$ | $\lambda_{\text{test}} = 20$ | $\lambda_{\text{test}} = 2$ | $\lambda_{\text{test}} = 20$ | $\lambda_{\text{test}} = (2, 20)$ |
| Latent ODE [6] | 3.826 | 5.935 | 3.066 | 3.027 | 2.690 |
| CTFP-IID-Gaussian | 4.952 | 4.094 | 3.025 | 3.024 | 2.716 |
| Latent CTFP-IID-Gaussian | 3.945 | 5.072 | 3.017 | 3.000 | 2.689 |
| CTFP (**ours**) | **3.107** | **1.929** | 2.902 | 1.941 | 1.408 |
| Latent CTFP (**ours**) | **3.107** | 1.930 | 2.902 | **1.939** | **1.392** |
| Ground Truth | 3.106 | 1.928 | 2.722 | 1.888 | 1.379 |

Table 3: Comparison Between CTFP, CTFP-IID-Gaussian, latent CTFP, and latent CTFP-IID-Gaussian on real-world datasets. We report NLL per observation.

| Model | Mujoco-Hopper [6] | BAQD [1] | PTBDB [7] |
|---|---|---|---|
| Latent ODE [6] | $24.775 \pm 0.010$ | $2.789 \pm 0.011$ | $-0.818 \pm 0.009$ |
| CTFP-IID-Gaussian | $22.023 \pm 0.010$ | $3.398 \pm 0.006$ | $-0.375 \pm 0.003$ |
| Latent CTFP-IID-Gaussian | $17.397 \pm 0.007$ | $1.471 \pm 0.005$ | $-1.436 \pm 0.005$ |
| CTFP (**ours**) | $-16.249 \pm 0.034$ | $-2.361 \pm 0.020$ | $-1.324 \pm 0.028$ |
| Latent CTFP (**ours**) | $\mathbf{-31.397 \pm 0.063}$ | $\mathbf{-6.894 \pm 0.046}$ | $\mathbf{-1.999 \pm 0.010}$ |

## C.2 I.I.D. Gaussian as Base Process

In this experiment, we replace the base Wiener process with I.I.D Gaussian random variables and keep the other components of the models unchanged. This model and its latent variant are named CTFP-IID-Gaussian and latent CTFP-IID-Gaussian. As a result, the trajectories sampled from CTFP-IID-Gaussian are not continuous and we use this experiment to study the continuous property of models and its impact on modeling irregular time series data with continuous dynamics. The results are presented in Table 2 and Table 3.

The results show that CTFP consistently outperforms CTFP-IID-Gaussian, and latent CTFP outperforms latent CTFP-IID-Gaussian. The results corroborate our hypothesis that the superior performance of CTFP models can be partially attributed to the continuous property of the model. Moreover, latent CTFP-IID-Gaussian shows similar but slightly better performance than latent ODE models. The results comply with our hypothesis as the models are very similar and both models have no notion of continuity in the decoder. We believe the performance gain of latent CTFP-IID-Gaussian comes from the use of (dynamic) normalizing flow which is more flexible than Gaussian distributions used by latent ODE.

## C.3 CTFP-RealNVP

In this experiment, we replace the continuous normalizing flow in CTFP model with another popular choice of normalizing flow model, RealNVP [3]. This is variant of CTFP is named CTFP-RealNVP and its latent version is called latent CTFP-RealNVP. Note that the trajectories sampled from CTFP-RealNVP model are still continuous. We evaluate CTFP-RealNVP and latent CTFP-RealNVP models on datasets with high dimensional data, Mujoco-Hopper, and BAQD. The results are shown in Table 4.

Table 4: Comparison between CTFP, CTFP-RealNVP, and their latent variants on Mujoco-Hopper and BAQD datasets. We report NLL per observation.

| Model | Mujoco | BAQD |
|---|---|---|
| CTFP-RealNVP | $-23.061 \pm 0.000$ | $-5.099 \pm 0.002$ |
| Latent CTFP-RealNVP | $-23.602 \pm 0.001$ | $-5.109 \pm 0.005$ |
| CTFP | $-16.249 \pm 0.034$ | $-2.361 \pm 0.020$ |
| Latent CTFP | $\mathbf{-31.397 \pm 0.063}$ | $\mathbf{-6.894 \pm 0.046}$ |

The table indicates that CTFP-RealNVP outperforms CTFP. However, when incorporating the latent variable, the latent CTFP-RealNVP performs significantly worse than latent CTFP. The worse

performance might be because RealNVP cannot make full use of the information in the latent variable due to its structural constraints as we discussed in Section 4.2.

## D    Additional Details for Latent ODE Models on Mujoco-Hopper Data

The original latent ODE paper focuses on point estimation and uses the mean squared error as the performance metric [6]. When applied to our problem setting and evaluated using the log-likelihood, the model performs unsatisfactorily. In Table 5, the first row shows the negative log-likelihood on the Mujoco-Hopper dataset. The inferior NLL of the original latent ODE is potentially caused by the use a fixed output variance of $10^{-6}$, which magnifies even a small reconstruction error.

To mitigate this issue, we propose two modified versions of the latent ODE model. For the first version (V1), given a pretrained (original) latent ODE model, we do a logarithmic scale search for the output variance and find the value that gives the best performance on the validation set. The second version (V2) uses an MLP to predict the output mean and variance. Both modified versions have much better performance than the original model, as shown in Table 5, rows 2–3. It also shows that the second version of the latent ODE model (V2) outperforms the first one (V1) on the Mujoco-Hopper dataset. Therefore, we use the second version (V2) for all the experiments in the main text.

Table 5: Comparison of different version of latent ODE models on Mujoco-Hopper Datasets.

| Model | NLL |
| --- | --- |
| Latent ODE (original) | $4 \times 10^7 \pm 9 \times 10^5$ |
| Latent ODE (V1) | $45.874 \pm 0.001$ |
| Latent ODE (V2) | $24.775 \pm 0.010$ |
| VRNN | $9.113 \pm 0.018$ |
| CTFP | $-16.249 \pm 0.034$ |
| Latent CTFP | $\mathbf{-31.397 \pm 0.063}$ |

## E    Qualitative Sample for VRNN Model

We sample trajectories from the VRNN model [2] trained on Geometric Brownian Motion (GBM) by running the model on a dense time grid and show the trajectories in Figure 1. We compare the trajectories sampled from the model with trajectories sampled from GBM. As we can see, the sampled trajectories from VRNN are not continuous in time.

(a) VRNN                    (b) Ground Truth

Figure 1: Sample trajectories and marginal density estimation by VRNN (a). We compare the results with sample trajectories and marginal density with ground truth (b). In addition to the sample trajectories (red) and the marginal density (blue), we also show the sample-based estimates (closed-form for ground truth) of the inter-quartile range (dark red) and mean (brown) of the marginal density.

We also use VRNN to estimate the marginal density of $\boldsymbol{X}_\tau$ for each $\tau \in (0, 5]$ and show the results in Figure 1. It is not straightforward to use VRNN model for marginal density estimation. For each

timestamp $\tau \in (0, 5]$, we get the marginal density of $\boldsymbol{X}_\tau$ by running VRNN on a time grid with two timestamps, 0 and $\tau$: at the first step, the input to VRNN model is $\boldsymbol{x}_0 = 1$ and we can get prior distributions of the latent variable $\boldsymbol{Z}_\tau$. Note that a sampled trajectory from GBM is always 1 when $\tau = 0$. Conditioned on the sampled latent codes $\boldsymbol{z}_0$ and $\boldsymbol{z}_\tau$, VRNN proposes $p(\boldsymbol{x}_\tau | \boldsymbol{x}_0, \boldsymbol{z}_\tau, \boldsymbol{z}_0)$ at the second step. We average the conditional density over 125 samples of $\boldsymbol{Z}_\tau$ and $\boldsymbol{Z}_0$ to estimate the marginal density.

The marginal density estimated using a time grid with two timestamps is not consistent with the trajectories sampled on a different dense time grid. The results indicate that the choice of time grid has a great impact on the distribution modeled by VRNN and the distributions modeled by VRNN on different time grids can be inconsistent. In contrast, our proposed CTFP models do not have such problems.

## Footnotes

*Work developed during an internship at Borealis AI. Correspond to wsdmdeng@gmail.com.