[Reviews · NeurIPS 2020]

Review 1

Summary and Contributions: This paper proposes using continuous-time neural ODEs to transform a latent Wiener process into a richer process over the observation space. The idea is similar to the usual formulation of normalizing flows, except that now the (usually fixed) base distribution is a stochastic process. Or to be more precise, the flow’s transformation warps the conditional Normal variables of the Brownian bridge. This conditional formation is used for exact likelihood calculation and sampling. Sampling the latent process via independent increments means that the observation process’ increments are independent as well. This is eventually relaxed by the addition of another latent variable. Experiments are first performed showing the ability to model synthetic data simulated from three different stochastic processes. Secondly, performance is demonstrated on continuous control, ECG modeling, and air quality prediction.

Strengths: Irregular sampling: The primary benefit of this model is its ability to handle irregularly sampled time series data. This paper’s transformation of just the increments allows for complete flexibility in the time steps and no RNN-based decoding, as previous work has required. (Yet note that, unlike for the Latent ODE, this work’s use of the Neural ODE has *nothing* to do with handing irregular sampling---more on this below.)

Weaknesses: No Explanation of Transformations of Stochastic Processes: I was under the impression that transforming / reparameterizing a stochasic process is non-trivial. I expected to see Ito’s lemma, which is analogous to the change of variables for SDEs: For Y_t = f(X_t), it’s derivative is: dY_t = f’(X_t) dX_t + ½ f’’(X_t) \sigma^2_t dt. Thus, I was expecting Equation 7 to include a second derivative term. Why doesn’t it? I’m not saying that Equation 7 is wrong, per se---transforming just the increments agrees with intuition. However, the problem is that the paper provides no explanation or mathematical references for stochastic processes and their transformations. There are *zero* citations in both Section 2.2 and Section 3.1. More work needs to be done to assure the reader that Equations 5-7 are correct, since they are the foundation of the work. Use of Neural ODEs: The paper says that Neural ODEs are used “because it has free-form Jacobian and efficient trace estimator” (line 158). It makes no sense to me why one would choose to encumber the model with such a slow, (relatively-)difficult-to-implement flow. Why not at least use a Residual Flow [Behrmann et al., 2019; Chen et al., 2019]? These are essentially the same except that now there’s no need to backprop through an ODE solver. Furthermore, the one ablation experiment done with an RNVP flow is not informative. As the paper mentions, the checkerboard masking of RNVP is very restrictive. Moreover, it is designed for images---a domain the paper does not consider. I suspect using Glow or an autoregressive flow (inversion cost not withstanding) would be competitive-to-superior to the Neural ODE-variant while being much more practical. Ablation study of sampling intervals: I would really have liked to have seen an ablation study of the sampling intervals. For the real-world data, the time stamps are sampled once from a homogeneous Poisson process with an intensity of 0.5 (line 36 of supp mats). Varying the sampling would have allowed us to see which of the models is truly best under irregular sampling. Perhaps Latent ODE and CTFP excel in different regimes and such a study would tell the reader which to use for their data. Repetitive Introduction: I find the Introduction to be quite repetitive. It could be shortened and the extra space used for the stochastic process background I mentioned above. Behrmann, Jens, et al. "Invertible residual networks." International Conference on Machine Learning. 2019. Chen, Ricky TQ, et al. "Residual flows for invertible generative modeling." Advances in Neural Information Processing Systems. 2019.

Correctness: The theoretical correctness is hard to verify because no exposition or even references are given for the transformation of stochastic processes.

Clarity: No. There are places for clear improvement. The Introduction is quite repetitive, and the stochastic process background material that underlies the method should be expanded.

Relation to Prior Work: I believe so but am not well-read in time series modeling.

Reproducibility: Yes

Additional Feedback: POST REBUTTAL UPDATE: Thank you, authors, for addressing my concerns. I have raised my score to a 6. While I agree that the paper's primary contribution is the flow's general formulation and the exact transformation is just an implementation choice, I still think it is important to report results for a wider range of transformation classes and to discuss trade-offs.


Review 2

Summary and Contributions: In this work, they use a normalizing flow to transform a Weiner process into a more complex stochastic process.

Strengths: Sound. Novel in the strict sense (I do not know of work proposing exactly this scheme).

Weaknesses: The key issue is in Eq. 6, which appears to indicate that the normalising flow transformation is applied at each timepoint independently. This renders the model somewhat trivial (a Kalman filter with nonlinear outputs), extensively studied (e.g. in the EKF literature). It is well known that linear dynamics in a Kalman filter is a discretisation of an underlying continuous dynamical system, and you could use unevenly spaced observations if desired. Here are some possible claims that would render the work more interesting: 1.) Inclusion of the Jacobean in the output transformation renders the ML solution a better characterisation of the Bayesian solution. 2.) Having a normalising flow that depends on the value of the Weiner process at all past time steps (which allows much richer temporal dependencies). 3.) Arguing that the presently described process is surprisingly effective, by looking at a more empirical examples, and comparing to SOTA performance from referenced papers. Eq 12 might do some of this, but it is extremely unclear.

Correctness: Technically reasonable.

Clarity: Well written.

Relation to Prior Work: The paper confuses "Background" and "Related Work". The Background should contain basic information e.g. on stochastic processes and normalising flows. The Related Work section should discuss a small number of the most closely related approaches (probably those that combine normalising flows with dynamical models), and tell us how the present work differs. Missing reference: Filtering Normalizing Flows (Razaghi 2019).

Reproducibility: Yes

Additional Feedback: While I cannot recommend acceptance at this stage, I hope that my suggestions will prove helpful for the authors.


Review 3

Summary and Contributions: The authors propose a new normalizing flow using a Wiener process, the continuous time flow process (CTFP). CTFP has many nice properties, such as being evaluated on irregular grids and having continuous sample paths. Although any normalizing flow model can be used under the authors’ framework, they use the ANODE model.

Strengths: The paper is very well written and easy to understand. The mathematical notation is also clear and precise. The related work section has the correct level of detail and explanation of previous work. The figures are extremely clear and useful. Figure 2 specifically was very helpful in visualizing the CTFP. The logic flow is excellent (no pun intended).

Weaknesses: Line 55 - The authors point out that continuity is important but do not explain why. Obviously this is a desirable property in general, but examples showing the limitations of non-continuous processes should be shown (or at least mentioned) to bolster the authors’ work. The authors claim that “the stochastic process generated by CTFP is guaranteed to have continuous sample paths, making it a natural fit for data with continuously-changing dynamics”; while this makes sense, it doesn’t quantify how important this is.

Correctness: Everything seems correct.

Clarity: The paper is extremely clear and well written. A few suggestions for improvements: Figure 1 is not referenced in the text, therefore it is easy to skip over. Equation 13 - The subscript of z in E_{\mathbf{z_k}∼q} is confusing. I am guessing that you are saying that each z_k is distributed according to q and that the expectation is over z1, ..., z_K. However, it is confusing because k is also used to index the sum inside of the expectation. Figure 3 should have labeled axes with a larger font size. Line 19 - The authors propose what a good time series model needs but doesn’t explain how this relates to their work or the work of others. Even adding something like “Our proposed model captures all of these properties while being computationally tractable” at Line 28 would help the reader understand that these are the properties that CTFP captures.

Relation to Prior Work: Yes previous contributions and the authors’ work are presented clearly.

Reproducibility: Yes

Additional Feedback: Overall the paper was very good. I would be happy to see it at NeurIPS.

[Author Response · NeurIPS 2020]

We would like to thank all the reviewers for their thoughtful comments. We will respond to each reviewer's questions
individually and incorporate the advice on formatting, notations, and references in an updated version of our manuscript.
**[R1] Explanation of the transformation.** Itô's Lemma shows our model can be used to construct a broad range of
Itô diffusion processes with tractable finite-dimensional distributions (FDD). To show the correctness of Eqs. (5-7),
it suffices to show the FDD on $\mathbb{R}^n$ of the stochastic process $\boldsymbol{X}_\tau$ defined by Eq.(6) is the same as the distribution
obtained by transforming the FDD of the Wiener process with the density of Eq.(7). We present a formal argument
based on measure theory and a proof sketch: consider the classical Wiener space $(\Omega, \Sigma)$, where $\Omega = C([0, +\infty), \mathbb{R})$,
the set of continuous functions from $[0, +\infty)$ to $\mathbb{R}$, and $\Sigma$ is the $\sigma$-algebra generated by all the cylinder sets of
$C([0, +\infty), \mathbb{R})$.[1] We can equip this space with a probability measure (distribution) $Q$ to get a probability space
$(\Omega, \Sigma, Q)$ for continuous-time stochastic processes. Given a finite subset $\{\tau_1, \tau_2, ..., \tau_n\} \subset (0, +\infty)$, define the
**projection** $\pi_{\{\tau_1,...,\tau_n\}} : (\Omega, \Sigma, Q) \longrightarrow (\mathbb{R}^n, \mathcal{B}(\mathbb{R}^n))$ to be $\pi_{\{\tau_1,...,\tau_n\}}(\omega) = (\omega(\tau_1), ..., \omega(\tau_n))$, where $\mathcal{B}(\cdot)$ is the
Borel $\sigma$-algebra. We will drop the index and simply use $\pi$ to denote projection from now on. Projection is a measurable
mapping. The **finite-dimensional distribution (FDD)** of a process is defined to be the pushforward measure induced
by $\pi$, that is $Q \circ \pi^{-1}(A) = Q(\{\omega : (\omega(\tau_1), ..., \omega(\tau_2)) \in A\}), A \in \mathcal{B}(\mathbb{R}^n)$. Let $P$ denote the unique measure of Wiener
process $\boldsymbol{W}_\tau$ defined on the classical Wiener space. The following proposition and theorem will serve our purpose:
**Proposition 1** *The mapping from* $(\Omega, \Sigma, P)$ *to* $(\Omega, \Sigma)$ *defined by* $\boldsymbol{X}_\tau = F_\theta(\boldsymbol{W}_\tau, \tau)$ *is measurable and therefore*
*induces a pushforward measure* $P \circ F_\theta^{-1}$.
**Theorem 1** *Given a finite subset* $\{\tau_1, \tau_2, ..., \tau_n\} \subset (0, +\infty)$, *the FDD of* $\boldsymbol{X}_\tau$ *is the same as the distribution of*
$(F_\theta(\boldsymbol{W}_{\tau_1}, \tau_1), ..., F_\theta(\boldsymbol{W}_{\tau_n}, \tau_n))$, *where* $(\boldsymbol{W}_{\tau_1}, ..., \boldsymbol{W}_{\tau_n})$ *is a n-dimensional random variable with FDD of* $\boldsymbol{W}_\tau$.
The distributions, or (pushforward) measures, of $(\boldsymbol{X}_{\tau_1}, ..., \boldsymbol{X}_{\tau_n})$ and $(F_\theta(\boldsymbol{W}_{\tau_1}, \tau_1), ..., F_\theta(\boldsymbol{W}_{\tau_n}, \tau_n))$ are induced by
two mappings from $(\Omega, \Sigma, P)$ to $(\mathbb{R}^n, \mathcal{B}(\mathbb{R}^n))$ respectively: a) $\pi \circ F_\theta$ and b) $F_\theta \circ \pi$, where $\pi$ is the projection onto
$\{\tau_1, \tau_2, ..., \tau_n\}$. To show the two distributions on $(\mathbb{R}^n, \mathcal{B}(\mathbb{R}^n))$ are equal, it suffices to check that they assign the same
measure to every Borel set of $\mathbb{R}^n$. This is true because the preimages of every Borel set under the two mappings are
identical. The arguments above can be generalized to $\Omega = C([0, +\infty), \mathbb{R}^n)$.
**[R1] Efficiency of neural ODE.** We would like to clarify that our main contribution is a continuous-time stochastic
process model, of which the normalizing flow (NF) is just one component. We use a continuous normalizing flow
(neural ODE) primarily because of its free-form Jacobian matrix property, flexibility w.r.t. transformations, and model
architecture. It also shows competitive performance on low-dimensional data compared with GLOW and autoregressive
flows. Since our experiments focus on low-dimensional data, the time cost is not a major bottleneck. Other types of
data may require a different choice of flow, which is possible and within the specifications of the proposed framework.
**[R1] Ablation study of sampling intervals.** It is worth noting that the observation time intervals are different samples
from the same Poisson process for each sequence. We would like to make a minor correction on data-preprocessing
   details: for real-world data we use $\lambda = 2$ rather than $\lambda = 0.5$. We present the requested ablation study results below:

Table 1: Ablation Study on Time Interval for Real-World Data

| Model | Mujoco-Hopper | | BAQD | | PTBDB | |
|---|---|---|---|---|---|---|
| | $\lambda_{\text{test}} = 1$ | $\lambda_{\text{test}} = 5$ | $\lambda_{\text{test}} = 1$ | $\lambda_{\text{test}} = 5$ | $\lambda_{\text{test}} = 1$ | $\lambda_{\text{test}} = 5$ |
| Latent ODE | $25.082 \pm 0.011$ | $24.599 \pm 0.004$ | $2.948 \pm 0.006$ | $2.686 \pm 0.006$ | $-0.633 \pm 0.006$ | $-0.892 \pm 0.009$ |
| VRNN | $10.553 \pm 0.010$ | $8.543 \pm 0.008$ | $0.044 \pm 0.007$ | $-1.016 \pm 0.001$ | $\mathbf{-1.552 \pm 0.011}$ | $\mathbf{-2.545 \pm 0.005}$ |
| CTFP | $-5.860 \pm 0.013$ | $-20.530 \pm 0.003$ | $-0.890 \pm 0.001$ | $-3.595 \pm 0.001$ | $-0.982 \pm 0.041$ | $-1.793 \pm 0.015$ |
| Latent CTFP | $\mathbf{-28.272 \pm 0.043}$ | $\mathbf{-32.388 \pm 0.057}$ | $\mathbf{-7.212 \pm 0.064}$ | $\mathbf{-6.157 \pm 0.035}$ | $-1.549 \pm 0.009$ | $-2.525 \pm 0.007$ |

**[R2] Independent normalizing flow transformation.** While there is similarity between the graphical model represen-
tations of our approach and state-space models, we would like to stress the fundamental differences between them:
CTFP directly models a distribution of continuous functions from the time axis to an observation space, or equivalently
a stochastic process. It takes the evaluations of the functions at an arbitrary given time-grid to be the distribution of
observations and can directly compute its density. The CTFP mapping is injective and does not rely on an emission
process with observation noise. We condition each transformation only on the time stamp $\tau$ rather than previous
observations to enforce marginalization consistency of the stochastic process: given the finite-dimensional distribution
(FDD) of $(\boldsymbol{X}_{\tau_1}, ..., \boldsymbol{X}_{\tau_i}, ..., \boldsymbol{X}_{\tau_n})$ for a stochastic process $\boldsymbol{X}_\tau$, the distribution obtained by marginalizing over one of
the dimensions, $\boldsymbol{X}_{\tau_i}$, must be the same as the FDD of $(\boldsymbol{X}_{\tau_1}, ..., \boldsymbol{X}_{\tau_{i-1}}, \boldsymbol{X}_{\tau_{i+1}}, ..., \boldsymbol{X}_{\tau_n})$. Our experiments show that
our models outperform VRNN, which conditions emission and transition on all previous observations and is arguably a
more powerful filtering-based model than the extended Kalman filter. We agree with the reviewer's comment on Eq.(12):
the latent variable $z$ could be interpreted as containing history information to relax the Markov property of CTFP.
**[R3] Importance of continuity.** We thank the reviewer for recognizing continuity as a unique property of CTFP. Its
importance is shown from two aspects in our work: our models show better performance than other models when
evaluated using denser observation intervals (larger $\lambda$) on most of the datasets. This is partially due to our model's
continuity as the dependence between neighboring observations is stronger with denser observations. Moreover, the
qualitative examples in Fig. 3(c) show that our models, which can generate continuous trajectories, are better for
interpolation than non-continuous models.

## Footnotes

[1] We refer the reviewer to Chapter 2 of *Brownian Motion, Martingales, and Stochastic Calculus* by Jean François Le Gall for more details.


[Meta-Review · NeurIPS 2020]

## Modeling Continuous Stochastic Processes with Dynamic Normalizing Flows: *PROS: handle irregularly sampled time series data. The paper is very well written and easy to understand. *CONS: paper confuses "Background" and "Related Work", lack of motivation for continuity. One reviewer recommend borderline rejection, but in my opinion the authors successfully addressed his concerns in the rebuttal. Recommendations: The authors are encouraged to clearly explain the reviewers' concern on potential similarities of the approach with the Kalman filter with nonlinear outputs. Also the issues related to background and related work and motivation for continuity.